# Effects of core stability exercises on balance ability of children and adolescents with intellectual disabilities: A systematic review and meta-analysis

**Junjie Zhou[ID], Yecheng Zhong[ID], Wenhong Xu[ID]***

College of Physical Education and Health Sciences, Zhejiang Normal University, Jinhua, Zhejiang Province, China

* xuwenhong@zjnu.edu.cn

## Abstract

### Background

Children and adolescents with intellectual disabilities (IDs) are at risk of falls due to balance problems. One way to palliate balance deficits among this population is via core stability exercises. However, comprehensive studies that examine the effectiveness of core stability exercises in improving balance in this target population are lacking.

### Objective

This study aims to summarise and quantify the effectiveness of core stability exercises in improving the balance of this target population.

### Methods

This study followed PRISMA principles and conducted comprehensive searches in six academic databases (PubMed, Web of Science, Medline, Embase, Scopus, and the Cochrane Library) up to June 2023. The inclusion criteria were established via the PICOS framework. The risk of bias was assessed with the Cochrane risk-of-bias tool, and the certainty of the evidence was assessed via the GRADE approach. The meta-analysis was performed via RevMan 5.4, and for data that could not be pooled via meta-analysis, we used a narrative description of the results of each study.

### Results

Six studies of 1078 subjects were included. The findings revealed that core stability exercises improved the dynamic balance of children and adolescents with ID but had no significant effect on static (Hedges' g = 1.32, 95% CI [-0.41 to 3.06]) or static–dynamic (Hedges' g = 1.35, 95% CI [-0.02 to 2.73]) balance compared with the control groups. The quality of evidence based on the GRADE approach was very low.

**Data Availability Statement:** All relevant data are within the manuscript and its Supporting Information files.

**Funding:** This work was supported by the Education Sciences Planning of Zhejiang Province, China [No. 2023SCG375 to WX], under the project titled "The assessment and physical activity intervention of children and adolescents with disabilities based on their 24-hour movement.

**Competing interests:** The authors declare that there is no conflict of interest.

## Conclusions

Core stability exercises may improve dynamic balance in children and adolescents with ID, but given the scarcity of studies included, definitive conclusions cannot yet be drawn. Although pooled analyses also highlighted improvements in static and static–dynamic balance with large effect sizes over active control groups, the results were not statistically significant and should be interpreted with caution given the wide confidence intervals. The heterogeneity among the identified studies and the limited number of eligible studies may reduce the reliability of the results, but these findings emphasise the need for additional research in this domain.

## Introduction

Balance is a prerequisite to learning complex motor skills during childhood [1] and the foundation of successful performance of daily and sport-related activities from childhood to adulthood [2]. Balance is the complex ability to maintain, achieve, or restore the state of equilibrium of the body while an individual stands still, prepares to move, is in motion, or prepares to stop moving [3,4]. This capability can be examined under static (the body remains motionless), dynamic (the body reacts to perturbations or is in movement), or both conditions [5]. Poor balance is a predictor of falling or a failure of results during an activity [6] and is always associated with low self-esteem and a sedentary lifestyle, which may lead to obesity, cardiovascular diseases, and a high risk of death from all causes [7,8]. Because balance performance gradually matures from the age of seven years and plateaus in early adolescence [9,10], attention should be given to the development characteristics and intervention efforts related to the balance ability of children and adolescents.

Intellectual disability (ID) is a condition characterised by significant limitations in both intellectual function and adaptive behaviour that originates before the age of 22 [11]. According to the American Association on Intellectual and Developmental Disabilities (AAIDD), people are classified as having an ID when their IQ score is less than 70 [11]. In addition to significant limitations in intellectual functioning and adaptive behaviour, children with ID are also characterised by delays in motor milestones and impairments in sensorimotor function that affect the sensory, neuromusculoskeletal and motor systems [12]. Moreover, individuals with IDs may have comorbid conditions, such as hearing impairments, visual impairments, epilepsy, and hypotonia, and these impairments may also affect their physical functioning [13,14]. With an incidence of approximately 2% of the general population [15], ID ranks among the top ten causes of disease burden globally [16]. IDs are often distinguished as genetic and nongenetic in origin, with Down syndrome (DS), which is caused by the trisomy of *Homo sapiens* chromosome 21, being the most common genomic ID disorder [17]. The prevalence of DS is generally higher in males than in females [18,19]. Children and adolescents with ID usually perform poorer than their typically developing peers do in fundamental movement skills [20,21], especially in balance performance [22]. Indeed, balance deficits among children and adolescents with IDs increase the risk of instability, falls, and fall-related injuries, which range from slight contusion to fractures or even death [5,23]. According to several Polish sources of epidemiological data, almost 40% of 9–17-year-old adolescents with IDs suffer from disturbed body statics [24]. Furthermore, falling has been suggested to be a frequent event for those with IDs [25] and accounts for 50–60.2% of injuries [26]. Because balance plays a role in providing stability to various movements in daily activities, improving the balance of children with IDs could reduce the incidence of fall accidents and improve their quality of life. [27,28].

One of the main considered components of balance is core stability [29,30], which is defined as "the ability to achieve and sustain control of the trunk region at rest and during precise movement" [31]. The balance state requires the coordinated work of the core muscles, where the centre of gravity of the body's anatomical position is located [32]. Both global (e.g., oblique abdominals and spinalis) or local stabilisers (e.g., multifidus, transverse abdominis, and the pelvic floor) [29] in the core significantly affect a person's posture by stabilising the pelvis and the spine and controlling postural fluctuations [33,34]. In biomechanical terms, physiological muscle activation results in several biomechanical effects that allow efficient local and distal functions. Preprogrammed muscle activations result in anticipatory postural adjustments, which position the body to withstand the perturbations in balance created by the forces of kicking, throwing, or running [30]. The anticipatory postural adjustments create proximal stability for distal mobility. Specifically, muscle activity of the trunk precedes dynamic movement of the extremities [35], suggesting that the abdominal muscles may be activated to provide a stable foundation for the production, absorption, and control of force and motion to the extremities. Consequently, insufficient core stability is believed to lead to less efficient movements and, ultimately, musculoskeletal injury [36]. In recent years, core stability exercises, such as those that activate specific motor patterns of the trunk muscles by challenging spinal stability and trunk postural control, have become common elements of training programs to improve balance ability in a wide range of populations, such as athletes [37], people with visual impairment [38], and children and adolescents with hearing impairment and cerebral palsy [39–41]. A systematic review and meta-analysis recently reported that core training improves the dynamic balance of athletes and the nontraining population, especially among young people [37]. Positive intervention effects on balance performance were also observed in children and adolescents with hearing impairment and cerebral palsy [39–41].

Despite the growing interest in using core stability exercises for children and adolescents with IDs, a comprehensive synthesis of their effects on balance improvement in this population is still lacking. This knowledge gap impedes our understanding of the potential benefits and feasibility of core stability exercises as a modality for improving balance in this population. Although some studies [42,43] have shown promising results for core stability exercises in children with IDs, the absence of systematic reviews that assess the efficacy of these exercises for different balance improvements is notable. Therefore, this systematic review and meta-analysis aimed to examine and synthesise the available evidence on the effects of core stability exercises on improving balance in this target population. By addressing this literature gap, this systematic review helps provide clinicians, researchers, and PE teachers with evidence-based insights to inform decision-making and enhance rehabilitation practices for further research in this crucial field.

## Methods

### Study protocol and registration

This review was registered with the International Prospective Register of Systematic Reviews (PROSPERO; Registration no. CRD42023439091; available from https://www.crd.york.ac.uk/PROSPERO/) and conducted and reported in accordance with the Preferred Reporting Items for Systematic Reviews and Meta-Analysis (PRISMA) statement [44].

### Information sources and search strategy

A systematic search of the following electronic databases was performed in June 2023: PubMed, the Web of Science Core Collection, Medline, Embase, Scopus, and the Cochrane Library. The search team used consistent search strategies. All the search strategies were

established by sharing among the research teams, and progress was assessed and shared during research meetings. Any discrepancies were resolved via discussion and agreement. The search terms were split across four facets: (1) intellectual disability, (2) children and adolescents, (3) core stability exercises, and (4) balance. Each of the search terms was derived from the literature. Additionally, the bibliographies of all included studies were manually checked for other potentially relevant publications. Additional details about the search strategy are available in S1 and S2 Tables.

## Inclusion and exclusion criteria

To be included in the current review, studies met the following criteria based on the Population Intervention Comparison Outcome Study design method [45]: (1) Population: Children and adolescents with IDs aged less than 18 years. All participants were diagnosed with ID using clearly defined or internationally recognised criteria; (2) Intervention: Core stability exercises should be used in the intervention periods; (3) Comparison: A standardised supervised conventional physical therapy program or any other training not focused on core exercising was used as an active control group; (4) Outcome: Effects of the intervention on the balance ability of participants as measured by a standardised tool were reported in detail; (5) Study design: Randomised controlled trial (RCT) or controlled trial.

Studies were excluded for the following reasons: (1) Studies of participants presenting with additional diagnostic criteria, such as cerebral palsy, attention deficit hyperactivity disorder, dyspraxia, epilepsy, physical disabilities, and visual impairments, were excluded because these additional cognitive, behavioural or physical conditions could influence participants' motor performance; (2) Case studies and nonoriginal studies (i.e., comments, reviews, and theoretical articles); (3) Studies lacking an intervention or in which core stability training was not the primary intervention; (4) Articles published in languages other than English; (5) Studies for which the full text or sufficient data could not be obtained.

## Study selection

After studies were identified through a database search, duplicate records were removed via EndNote X9. Two independent reviewers (JJZ and YCZ) screened the titles and abstracts of all the records retrieved using the same selection criteria, and full-text articles were obtained for potentially relevant records. During this phase, any articles whose full texts could not be accessed were excluded. Agreement between reviewers was required for the inclusion of full-text articles. Discrepancies were resolved by discussions with a third reviewer (WHX).

## Data extraction

Two researchers (JJZ and YCZ) independently extracted and cross-checked the data. Discrepancies were resolved through a consensus discussion. Similarly, a third author (WHX) was contacted to make the final decision if any disagreement persisted. A calibration exercise before starting the review was conducted to ensure consistency across reviewers. All extracted data were transferred into a standardised Microsoft Excel 2016 spreadsheet (Microsoft Corporation). The data extracted from the eligible articles included the following: (1) Study details: the first author, the publication year, the geographic location, and the recruitment setting; (2) Participant characteristics, e.g., age, sex, sample size, and ID level; (3) Characteristics of the intervention and control groups, e.g., sample size, percentage of males, age range, research designs, intervention content, total duration, frequency, and volume; and (4) Outcome details, such as the means and standard deviations of the balance outcome measurements and the names of the balance tests/s. In the case of missing data, the corresponding author of the

respective article was contacted by e-mail. If the author did not reply to the inquiry, the data were considered irretrievable.

## Risk of bias assessment

The Cochrane Collaboration's risk-of-bias tool was used to evaluate the included RCTs [46]. This tool is valid and reliable and consists of seven domains (i.e., random sequence generation, allocation concealment, blinding of participants and personnel, blinding of outcome assessment, incomplete outcome data, selective reporting, and other potential sources of bias) [46]. The appraisal of these domains in each RCT was assessed as having either a high, low, or unclear risk of bias. The first and second authors independently assessed the risk of bias in the included studies. If a consensus could not be reached, an agreement was reached via a discussion with the third author.

## Certainty of evidence

The certainty of the evidence was evaluated by two authors (JJZ and YCZ) via the Grading of Recommendations, Assessment, Development and Evaluation (GRADE) methodology, which categorised it as very low, low, moderate, or high [47]. The GRADE criteria included the following: (1) limitations of study design (risk of bias across trials); (2) indirectness of evidence; (3) inconsistency in results; (4) imprecision of results; and (5) risk of publication bias. Each outcome was categorised as "no", "serious" or "very serious". Grade-Pro software was used to evaluate the certainty of the evidence indicators in the literature. If the number of comparison trials was insufficient to conduct a meta-analysis, the evidence was automatically considered to be of very low certainty [48]. Therefore, for the outcomes not included in the meta-analyses, the certainty of evidence should be regarded as very low [49].

## Statistical analysis and synthesis

The studies in this review included the effects of core stability exercises on different balance abilities (i.e., static balance, dynamic balance, and static–dynamic balance). If at least two identical outcome variables (e.g., standing time) of the same task (e.g., one leg stance) were available, meta-analysis was performed via Review Manager software (Version 5.4.1; The Nordic Cochrane Centre, The Cochrane Collaboration, Copenhagen, Denmark). We used the standardised mean difference (SMD) with a 95% confidence interval (CI) to analyse the effect size. If statistical heterogeneity was identified across studies ($I^2 \geq 50\%$, $p < 0.10$), we applied the random-effects model. Otherwise, the fixed-effects model was applied. Hedges' g method was used to reflect the magnitude of the intervention effect [50,51]. According to Cohen et al. [52], a small effect is between $\geq 0.2$ and $< 0.5$, a medium effect is between $\geq 0.5$ and $< 0.8$, and a large effect is $\geq 0.8$. Notably, when a study contained two or more control groups, each result was separately analysed. For data that could not be synthesised using meta-analyses, we adopted a narrative description of the results across studies.

## Results

### Search results

A total of 184 studies were identified in the systematic literature search (Web of Science Core Collection, $n = 44$; Scopus, $n = 61$; Medline, $n = 1$; PubMed, $n = 17$; Cochrane Library, $n = 49$; and Embase, $n = 12$). These studies were exported to EndNote X9, and 59 duplicates were eliminated. The titles and abstracts of the remaining 125 articles were then screened, resulting in the exclusion of 115 studies. In total, the full texts of 10 studies were reviewed, six of which

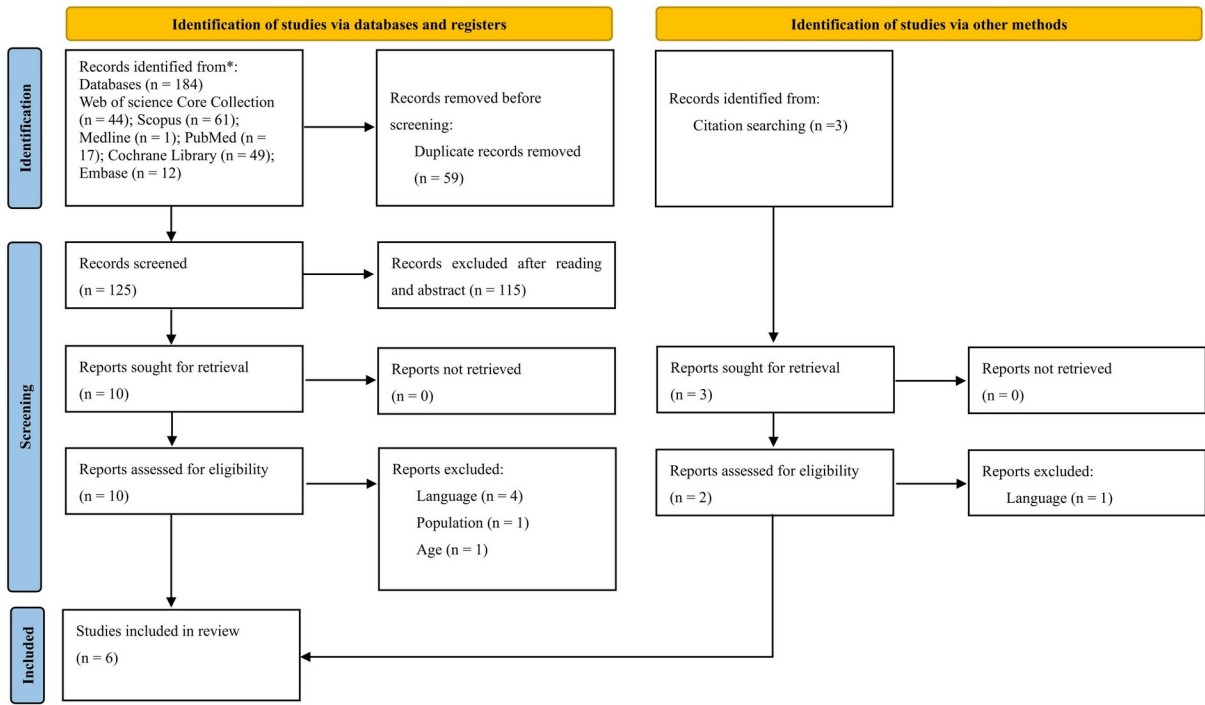

**Fig 1. The PRISMA flow diagram of the search and screening process.**

were excluded according to the eligibility criteria (S1 Appendix). Two additional studies were identified by checking the reference lists of the included studies. Ultimately, six primary studies were included. The PRISMA flowchart shows the selection procedure (Fig 1).

## Risk of bias in the included studies

Fig 2 shows the evaluation of the risk of bias in six RCTs. Three studies [53–55] described how random sequences were generated (e.g., computer randomisation and the lottery method), and the remaining studies [42,43,56] mentioned "random" but did not explain the specific method they adopted. One study [54] presented a low risk of bias because it reported randomisation masking, whereas five studies [42,43,53,55,56] presented an unclear or high risk of bias because they recorded inadequate information about the methods used for allocation concealment. The subjects and personnel were not blinded due to the particularity of the intervention approaches, which resulted in a high risk of performance bias in the included trials. One study [55] reported the blinding of outcome assessors and was rated as low risk, whereas the remaining studies [42,43,53,54,56] had an unclear risk of bias. All the studies presented a low risk of attrition bias because they reported all the data from the study sample. Five studies [42,43,53,55,56] presented a low risk of selective reporting because they included all the measured outcomes, and in one study [54], the presence of selective reporting was unclear.

## Study characteristics

As illustrated in Table 1, three studies (50%) were conducted in Iran, followed by Egypt ($n = 2$, 33.3%) and Pakistan ($n = 1$, 16.7%). A total of 173 participants were recorded, with chronological ages varying on average from 7 years to 11 years. The participants were recruited from outpatient clinics or rehabilitation centres ($n = 4$, 66.7%). Three studies [42,43,53] reported the

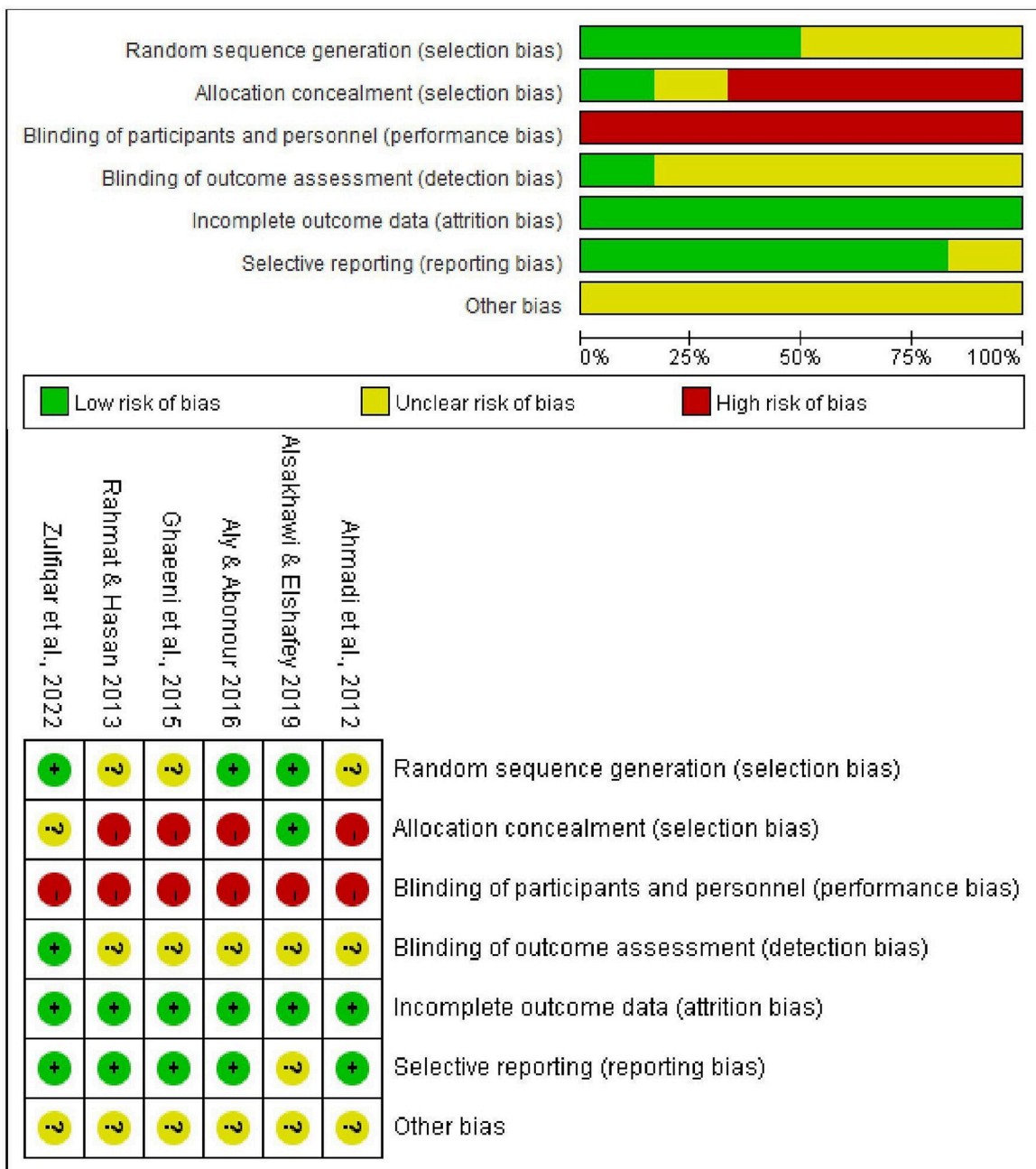

**Fig 2. Risk of bias graph and summary.**

participants' sex, and most participants were male. In four of the six studies [53–56], children with DS were recruited. In addition, on the basis of the Diagnostic and Statistical Manual of Mental Disorders (4th edition) [57] classification criteria (mild: IQ level 50–55 to approximately 70; moderate: IQ level 35–40 to 50–55; severe: IQ level 20–25 to 35–40; and profound: IQ level below 20 or 25), only three studies [42,43,53] reported ID severity (ranging from mild to moderate).

As illustrated in Table 2, all the studies adopted the RCT design. Core stability exercises based on the Jeffrey protocol were the most commonly used exercises in the six articles

Table 1. Characteristics of the population in the included studies (*n* = 6).

| Study | Country | Recruitment Setting | Age range (years) | ID level | Intervention Group | | Control Group | |
|---|---|---|---|---|---|---|---|---|
| | | | | | Sample Size (N) | % of Girls | Sample Size (N) | % of Girls |
| Ahmadi et al. 2012 [42] | Iran | NM | I: 11.23±1.95 C: 11.07±3.02 | Mild | 17 | 0 | 14 | 0 |
| Alsakhawi & Elshafey, 2019* [54] | Egypt | Outpatient clinic | 4 to 6 | NM | 15 | NM | C1: 15 C2: 15 | C1: NM C2: NM |
| Aly & Abonour, 2016* [53] | Egypt | Outpatient clinic | 6 to 10 | Mild-Mod | 15 | 26.7 | 15 | 33.3 |
| Ghaeeni et al. 2015* [56] | Iran | Rehabilitation center | 8 to 13 | NM | 8 | NM | 8 | NM |
| Rahmat & Hasan, 2013 [43] | Iran | NM | I: 11.23±1.95 C: 11.07±3.02 | Mild | 17 | 0 | 14 | 0 |
| Zulfiqar et al. 2022* [55] | Pakistan | Rehabilitation center | 5 to 17 | NM | 10 | NM | 10 | NM |

NM, not mentioned; Mod, moderate; I: intervention Group; C, control group; C1, control group 1; C2, control group 2

*, Participants with Down syndrome.

included. Except for three studies [42,43,55] that did not report the duration of each exercise, the length of the core stability exercises ranged from 30 min to 60 min per session, with three to four sessions per week and six to eight weeks in total. Five studies [42,43,53–55] presented detailed control conditions, and one did not [56]. Finally, none of the studies included a follow-up period.

Two [43,56] studies assessed static balance via the modified stock test (MST) and the one-leg stance (OLS) test. Three studies [42,53,54] assessed dynamic balance via the Y balance test (Y TEST) or the Biodex balance system (BBS). Two studies [54,55] assessed static–dynamic balance via the Berg balance test (BBT) or the Paediatric Berg scale (PBS). In these studies, the measures of balance included standing time (in seconds) (*n* = 2), stability index scores (overall, anterior-posterior, and mediolateral) (*n* = 2), limits of stability (anterior, posteromedial, and posterolateral) (*n* = 1), and the balance scale score (*n* = 2).

## Effects of core stability exercises on balance ability

According to different outcome indicators, the included studies were divided into three groups for analysis (i.e., the static, dynamic, and static–dynamic balance groups). The overall results from the studies are presented in Table 3. Compared with the control groups, the core stability exercise groups presented significantly greater posttest values for dynamic, static, and static–dynamic balance. Within the exercise group, these studies reported a significant improvement (from pretest to posttest) in three groups of balance.

**Dynamic balance.** Because different balance parameters were used, we narratively describe the results of the dynamic balance [42,53,54]. Two studies [53,54] that used the Biodex balance system concluded that core stability exercises effectively improved the overall stability index scores in children and adolescents with DS. Additionally, one study [42] used the Y balance test as a measurement tool and reported that participants in the intervention group had a significant improvement in mean posterolateral and posteromedial excursion in the posttest assessment($p < 0.005$), but that in the anterior direction was insignificant ($p = 0.271$).

**Static balance.** Static balance was assessed in two RCTs [43,56], and the data were pooled (Fig 3). The one leg stance test and the modified stork test were used to measure the static balance of 47 participants with IDs. The overall pooled effect size of the static balance function in the experimental group did not significantly differ from that in the control group [Hedges' g:

**Table 2. Characteristics of intervention programs, control conditions, and balance measures in the included studies (*n* = 6).**

| Study | Study design | Intervention | | | | Control condition | Balance measures | | |
| --- | --- | --- | --- | --- | --- | --- | --- | --- | --- |
| | | Content | Total duration (weeks) | Frequency (times/ week) | Volume | | Types | Measure | Parameters |
| Ahmadi et al., 2012* [42] | RCT | Abdominal crunch, back extension, hip raise and Russian twist on a stability ball; supine opposite arm-leg raises | 6 | 4 | Progressive increase in exercise | Training routines | Dynamic balance | Y TEST | Anterior, Posterolateral and Posteromedial excursion |
| Alsakhawi & Elshafey, 2019 [54] | RCT | 30mins traditional exercise program + 30mins core stability exercises based on Jeffrey protocol | 8 | 3 | Progressive increase in exercise | C1: 60mins traditional exercise program; C2: 5mins active stretching exercises+30mins traditional exercise program+20mins treadmill exercise program + 5mins cooldown | Static-dynamic balance; Dynamic balance | BBT; BBS | Total Test score; Overall Stability Index |
| Aly & Abonour, 2016* [53] | RCT | 45-60mins core stability exercises based on Jeffrey protocol | 8 | 3 | Progressive increase in exercise | Conventional physical therapy program | Dynamic balance | BBS | Overall, anterior-posterior and medial-lateral stability index scores |
| Ghaeeni et al., 2015 [56] | RCT | 45-60mins core stability exercises based on Jeffrey protocol | 8 | 3 | Progressive increase in exercise | NM | Static balance | MST | Standing time(s) |
| Rahmat & Hasan, 2013 [43] | RCT | Abdominal crunch, back extension, hip raise and Russian twist on a stability ball; supine opposite arm-leg raises | 6 | 4 | Progressive increase in exercise | Training routines | Static balance | OLS | Standing time(s) |
| Zulfiqar et al., 2022 [55] | RCT | Abdominal bracing with or without heel slide in lying position, or leg lifts with same position, same abdominal position with bridging, or standing with abdominal bracing | 6 | 3 | NM | Trunk balance exercises | Static-dynamic balance | PBS | Total Test score |

NM, not mentioned; RCT, randomised controlled trial; C, control group; C1, control group 1; C2, control group 2; Y TEST, Y balance test; OLS, one leg stance; MST, modified stork test; BBS, Biodex balance system; BBT, Berg balance test (scale); PBS, Paediatric berg scale

*, Articles not included in meta-analysis.

1.32 (95% CI: -0.41, 3.06) at $p$ = 0.13], with a highly significant difference in heterogeneity ($I^2$ = 81% at $p$ = 0.02).

**Static–dynamic balance.** Two studies used the Berg balance test and Paediatric Berg scale to investigate participants' static–dynamic balance ability [54,55]. We analysed 3 effect sizes for one study with two control groups (Fig 4). The overall pooled effect size of the static–dynamic balance function in the experimental group was not significantly greater than that in the control group [Hedges' g: 1.35 (95% CI: -0.02, 2.73) at $p$ = 0.05]. $I^2$ indicated a high level of heterogeneity ($I^2$ = 86% at $p$ < 0.001).

## Certainty of evidence

Because of the risk of bias, inconsistency, and imprecision, the level of evidence was down-graded. The strength of the evidence was not upgraded because of the limitations in the design

**Table 3. Results reported for outcomes of included studies (*n* = 6).**

| Study | IG (Pretest vs Posttest) | CG (Pretest vs Posttest) | IG vs CG (Posttest) | IG vs CG (Pretest-Posttest Change) |
|---|---|---|---|---|
| Ahmadi et al., 2012 [42] | A-E (*d* = 1.17)<br>PL-E (*d* = 1.07) **<br>PM-E (*d* = 1.49) ** | A-E (*d* = 0.78)<br>PL-E (*d* = 1.03)<br>PM-E (*d* = 0.33) | A-E (*d* = 0.65) **<br>PL-E (*d* = 2.11) **<br>PM-E (*d* = 1.07) ** | NM |
| Alsakhawi & Elshafey, 2019 [54] | Total (*d* = 5.09) **<br>Overall (*d* = 5.80) ** | C1: Total (*d* = 1.69) **<br>Overall (*d* = 2.10) **<br>C2: Total (*d* = 3.89) **<br>Overall (*d* = 7.42) ** | IG > CG1: Total (*d* = 2.96) *<br>Overall (*d* = 3.20) *<br>IG vs CG2: Total (*d* = 0.47)<br>Overall (*d* = 1.12) | NM |
| Aly & Abonour, 2016 [53] | A-P (*d* = 4.26) **<br>M-L (*d* = 3.04) **<br>Overall (*d* = 4.09) ** | A-P (*d* = 2.58) **<br>M-L (*d* = 1.80) **<br>Overall (*d* = 1.37) ** | A-P (*d* = 1.65) **<br>M-L (*d* = 1.23) **<br>Overall (*d* = 1.57) ** | NM |
| Ghaeeni et al., 2015 [56] | Standing time (*d* = 2.11) ** | Standing time (*d* = 0.07) | Standing time (*d* = 2.44) ** | NM |
| Rahmat & Hasan, 2013 [43] | Standing time (*d* = 0.60) * | Standing time (*d* = 0.41) | Standing time (*d* = 0.51) * | NM |
| Zulfiqar et al., 2022 [55] | Total (*d* = 1.30) * | Total (*d* = 0.46) | Total (*d* = 0.88) * | NM |

A-E, Anterior excursion; A-P, Antero-posterior stability indices; M-L, Medio-lateral stability indices; Overall, Overall stability indices; PL-E, Posterolateral excursion; PM-E, Posteromedial excursion; Total, Total Test score; CG, control group; CG1, control group 1; CG2, control group 2; IG, intervention group; NM, not mentioned

*, $p < 0.05$

**, $p < 0.01$.

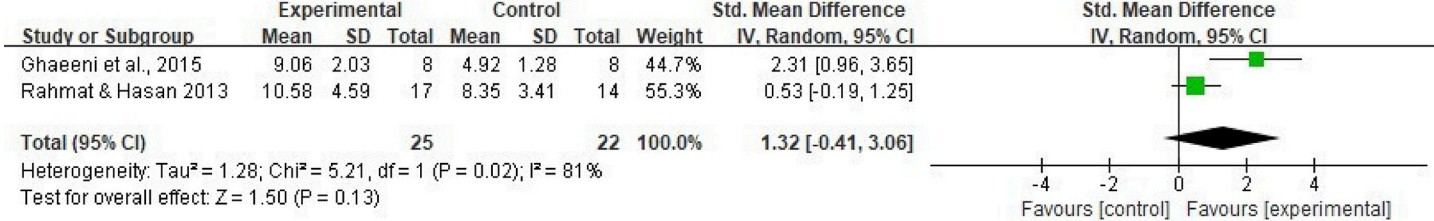

**Fig 3. Forest plot for the effect of core stability exercise on static balance.**

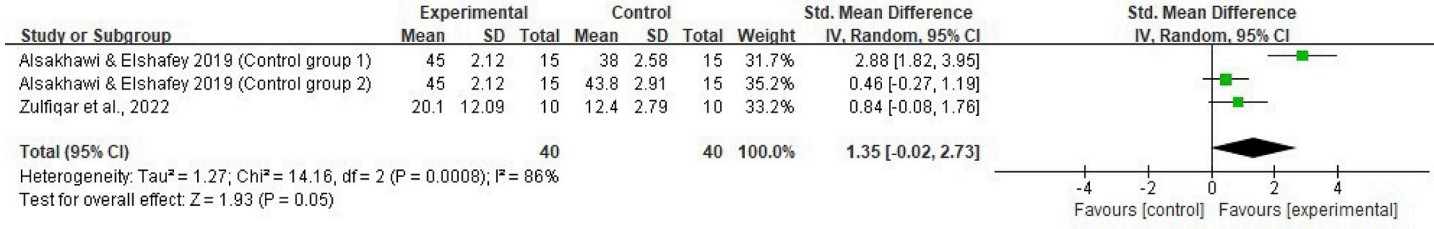

**Fig 4. Forest plot for the effect of core stability exercise on static–dynamic balance.**

of the included studies. Finally, the results revealed that the quality of the evidence (GRADE) was very low (Table 4).

## Discussion

This systematic review and meta-analysis aimed to assess the effects of core stability exercises on the balance ability of children and adolescents with IDs. The results of our review show that

**Table 4. Certainty of evidence for meta-analysed outcomes.**

| Outcome | Certainty assessment | | | | | | No of patients | | Effect | Certainty |
|---|---|---|---|---|---|---|---|---|---|---|
| | No of studies | Study design | Risk of bias | Inconsistency | Indirectness | Imprecision | Other considerations | Intervention group | Control group | Absolute (95% CI) | |
| **Static Balance** | 2 | RCT | Serious[1] | Very serious[2] | None | Serious[3] | None | 25 | 22 | SMD 1.32 (0.41 lower to 3.06 higher) | ⊕⊖⊖ **Very low** |
| **Static-dynamic balance** | 3 | RCT | Serious[1] | Very serious[2] | None | Serious[3] | None | 40 | 40 | SMD 1.35 (0.02 lower to 2.73 higher) | ⊕⊖⊖ **Very low** |

GRADE Working Group grades of evidence: **High quality:** Further research is very unlikely to change our confidence in the estimate of effect; **Moderate quality:** Further research is likely to have an important impact on our confidence in the estimate of effect and may change the estimate; **Low quality:** Further research is very likely to have an important impact on our confidence in the estimate of effect and is likely to change the estimate; **Very low quality:** We are very uncertain about the estimate.

[1] there is a high risk of bias across trials for at least one important domain, and it is unclear for one or more domains. Specifically, there was a high risk of bias for the blinding of participants and personnel (performance bias) and selection bias.

[2] there is considerable heterogeneity in the studies.

[3] the included studies recorded a small sample size for both the control and intervention groups, while the 95% CI crossed the line of no effect and the 95% CI was relatively wide.

core stability exercises significantly improved dynamic balance in this target population (Table 3). For static and static–dynamic balance, core stability exercises tend to be effective in individuals with IDs. However, the effects are insignificant compared with the effects on peers in the control groups (Figs 3 and 4).

Dynamic balance refers to a subject's ability to react efficiently to the base of support displacement [58]. The present review suggests that core stability exercises are beneficial for dynamic balance ability in children and adolescents with IDs, which is consistent with the findings of review that examined the effects of core stability exercises on athletes and non-trained populations [37]. Given the limits of intellectual function, children and adolescents with IDs usually experience delays in cognitive development and motor function [59]. Many studies have shown that sedentary lifestyles and obesity are highly prevalent in children and youth with IDs compared with typically developing peers [60,61], which in turn further hinders physical activity participation and balance development. According to Anderson et al. [32], interaction between central anticipatory and reflexive actions results in balance, and these actions are assisted by the active and passive restraints caused by the muscular system. Factors that influence balance include the sensory information obtained from the somatosensory, visual, and vestibular systems and motor responses that affect coordination, the joint range of motion, and strength [62]. Repetitive training experiences that influence motor responses and the earlier activation of core muscles by the central nervous system in anticipation of movements may be the underlying mechanism of balance improvement [37]. Moreover, complex environmental changes (e.g., visual or motor) during training may be beneficial for sensorimotor systems in children and adolescents with IDs. Additionally, core stability exercises can improve hip and trunk muscle strength, which helps build a solid, stable base and is important for increasing dynamic balance, especially in children and youth with IDs who exhibit relatively weak muscle strength [63]. The maintenance and recovery of balance is possible because of the coordinated work of muscles in the core, where the centre of gravity of the body's anatomical position is located [30,32]. The exercise most commonly used in these studies is the abdominal crunch, which recruits deep muscles of the spine, such as the lumbar

multifidus, and abdominal muscles, such as the transversus abdominis [64]. These deep muscles of the trunk support the spine and maintain the dynamic balance of the lower extremities [30,53].

The results of this review show that core stability exercises had a positive effect on dynamic balance in participants with a mean age of 7–11 years. Most of the study populations consisted of children, possibly because researchers focused more on the most critical period of motor development, that is, the first decade of life [65,66]. However, plateaus of balance are usually observed in early adolescence (i.e., 12–14 years), at which balance ability is largely equivalent to that of the average adult [10,67]. That is, balance still needs attention in the teenage years [68]. During the adolescent growth spurt, a differentiation exists between fast bone growth and low growth of musculature, coupled with a decrease in tendons flexibility [69]. For adolescents with IDs, increasing body dimensions and changing proportions can be a challenge when they are shaping skills to maintain body balance [70]. Therefore, future studies should pay more attention to adolescents with IDs to observe the results of interventions aimed at improving balance ability.

The present review also revealed that a common feature of the interventions was the progressive increase in the volume of core stability exercises. The progression of the exercise training load is one of the main training principles in exercise intervention [71]. A previous study conducted by Vera-Garcia et al. [72] revealed that the progression of exercises based on an individual's development and the movement pattern learned seem to be more decisive for dynamic balance performance. In the present review, three studies [42,53,54] on improved dynamic balance increased the volume progressively during the core stability exercises. Two of these studies [53,54] used the Jeffery protocol [73], which consists of exercises of progressively increasing difficulty that focus on strengthening the abdominal, lower back, and pelvic muscles. This result is consistent with previous findings in different populations, which showed that dynamic balance increased after progressive exercise [37,74,75].

In terms of the instruments used to evaluate outcomes, two studies [53,54] that used the Biodex balance system concluded that core stability exercises effectively improved the overall stability index score in children and adolescents with DS. The overall stability index score is believed to be the best indicator of the overall ability to balance the platform [76]. As a tool for measuring dynamic balance, Ahmadi et al. [42] used the Y test to assess three types of movements, namely, anterior, posteromedial, and posterolateral movements. The lower quarter Y test (which used only 3 axes of the star excursion balance test: anterior, posteromedial, and posterolateral) is a common method to assess dynamic balance [37]. Interestingly, core stability exercise improved posteromedial and posterolateral excursion but had insignificant effects in the anterior direction. Electromyographic activity in the lower limbs has been reported to depend on direction in the star offset balance test, with greater activation in certain directions than in others [77]. This finding indicates that core stability exercises have few effects in the anterior direction for neuromuscular system control and muscle strengthening [42].

The quantitative results revealed that core stability exercises had no significant effect on static ($p = 0.13$) or static–dynamic ($p = 0.05$) balance in participants with IDs compared with the control group. Moreover, significant heterogeneity existed in the results of both of our meta-analyses. This outcome could be caused by three factors. First, the number of studies and participants included in the analysis was relatively small. Second, the characteristics of the participants may have varied. Lipowicz et al. [70] noted that moderate disability is a factor that significantly reduces the ability of children and youth with ID to maintain balance. However, among the included studies, only one [43] reported the ID severity of the participants. Third, although most of the core stability exercises consisted of unified training content, the included research may cover inconsistent training approaches to conduct interventions, such as the

instruction technique or supervision process. Given the limited number of studies and their heterogeneity, more research is needed in the future to precisely estimate the effectiveness of core stability exercises on static and static–dynamic balance.

## Certainty of evidence

The overall certainty of evidence contributing to the meta-analysis was very low, suggesting that the current findings should be interpreted with caution. We could not perform an analysis restricted to high-quality trials because available trials at low risk of bias are insufficient for such an analysis. The main reason for the downgrading of the evaluation was study limitations, mainly in that the implementation of the blinding and allocation concealment schemes in the original study was imperfect. This limitation may be due to the characteristics of the intervention, which makes blinding the patients and investigators difficult. Both trial-level selection and performance bias have been shown to impact the size of the effect estimate [78,79]. The second reason for the downgrading is the high degree of heterogeneity among the original studies and the small sample sizes of the trials, which directly reduce the reliability of the evidence. In addition, the confidence intervals for the outcomes were wide, which resulted in imprecision. The above factors can lead to discrepancies between study findings and the actual situation. Therefore, more high-quality randomised controlled studies are needed to provide reliable data [80].

## Limitations

Among the six selected studies, five [42,43,53,54,56] were conducted in the Eastern Mediterranean region. Thus, the findings may not be generalisable to other nations and ethnicities. Another limitation is the probability of publication bias, which we attempted to decrease via a substantial database search. However, only studies published in English were included due to language barriers and resource limitations. Studies published in national languages in Asia or other countries were not included in this review. Finally, a limited number of studies were included in this review, and despite their similar characteristics, they remain heterogeneous in their interventions, control conditions, and characteristics of the participants.

## Implications

Despite these limitations, implications for future research can be derived from the present review. First, although the review studies were RCTs, most of the studies failed to demonstrate concealed allocation or the blinding of participants and assessors, which are all required to substantiate evidence of the strong efficacy of an intervention [81]. Moreover, the included studies lacked strict study designs and adequate sample sizes to optimise the power of the statistical analysis, which could affect the maturity of the measured outcomes. Second, core stability exercises were only conducted for a short period (6–8 weeks), which is insufficient to support the physical and cognitive integration of new skills to modify balance ability in children and adolescents with IDs in the long term [82]. Planning an appropriate duration to promote the full realisation of each child's balance potential may be more helpful. Third, future studies need to provide sufficient details about their exercise interventions, such as their settings, the facilitator (i.e., the person who delivers the intervention), and the adaptation of the intervention during the study.

## Conclusions

This review indicates that core stability exercises may be a promising means to improve dynamic balance in children and adolescents with IDs, which suggests that incorporating

them as an important part of physical and social rehabilitation programs is feasible for these children. In addition, although the quantitative results highlight improvements over active control groups in static and static–dynamic balance with large effect sizes, the results were not statistically significant and should be interpreted with caution because of the wide confidence intervals. Therefore, further rigorous studies are needed to strengthen the evidence in this area given the low quality of the underlying evidence base.

## Supporting information

**S1 Table. Details of search query.**
(DOCX)

**S2 Table. PRISMA 2020 Checklist.**
(DOC)

**S1 Appendix. List of excluded studies.**
(DOCX)

## Acknowledgments

The authors are grateful to all the authors' contribution during the completion of this study.

## Author Contributions

**Conceptualization:** Wenhong Xu.

**Data curation:** Yecheng Zhong.

**Formal analysis:** Junjie Zhou, Wenhong Xu.

**Methodology:** Junjie Zhou, Yecheng Zhong.

**Software:** Junjie Zhou.

**Supervision:** Wenhong Xu.

**Writing – original draft:** Junjie Zhou.

**Writing – review & editing:** Wenhong Xu.

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
