## [Decision Letter · Decision Letter 0]

16 Jul 2024

PONE-D-24-21189Effects of core stability exercises on balance ability of children and adolescents with intellectual disabilities: A systematic review and meta-analysisPLOS ONE

Dear Dr. xu,

Thank you for submitting your manuscript to PLOS ONE. After careful consideration, we feel that it has merit but does not fully meet PLOS ONE’s publication criteria as it currently stands. Therefore, we invite you to submit a revised version of the manuscript that addresses the points raised during the review process.

We look forward to receiving your revised manuscript.

Kind regards,

Domiziano Tarantino, MD

Academic Editor

PLOS ONE

Journal Requirements:

Reviewers' comments:

Reviewer's Responses to Questions

**Comments to the Author**

1. Is the manuscript technically sound, and do the data support the conclusions?

Reviewer #1: Partly

Reviewer #2: Partly

Reviewer #3: No

Reviewer #4: Yes

Reviewer #5: Partly

Reviewer #6: Yes

2. Has the statistical analysis been performed appropriately and rigorously? 

Reviewer #1: I Don't Know

Reviewer #2: Yes

Reviewer #3: N/A

Reviewer #4: Yes

Reviewer #5: Yes

Reviewer #6: Yes

3. Have the authors made all data underlying the findings in their manuscript fully available?

Reviewer #1: Yes

Reviewer #2: Yes

Reviewer #3: No

Reviewer #4: Yes

Reviewer #5: Yes

Reviewer #6: Yes

4. Is the manuscript presented in an intelligible fashion and written in standard English?

Reviewer #1: No

Reviewer #2: Yes

Reviewer #3: No

Reviewer #4: Yes

Reviewer #5: Yes

Reviewer #6: Yes

5. Review Comments to the Author

Reviewer #1: I think it is a different field that can be reviewed. but I do not find it appropriate to publish it because I think that the existing studies are not sufficient for meta-analysis and the results will be misleading.

Reviewer #2: In this manuscript, the authors summarise the available evidence on the effectiveness of core stability exercises in improving balance in children and adolescents with intellectual disabilities. The topic is clinically relevant and the results emphasise the need for more primary studies on the subject. However, some questions need to be properly addressed:

Abstract

Line 39/41- “Core stability exercises are effective in improving dynamic balance in children and adolescents with ID. However, no significant effects were found on the improvement in static and static-dynamic balance in this population”. Please consider clarifying the content of this sentence adding information about heterogeneity in the included sources of evidence, and the lack of significant improvements in comparison with the control group.

Methods

Line 126/127- According to the inclusion criteria, the average age of the participants should be < 18 years. However, the mean is a measure of central location, not indicate sample extremes. For example, in a hypothetical sample of 2 participants, if one is 30 years old and the other is 2 years old, the average will be 16 years old.

Line 128/129- The comparator is not properly described. The review aims to assess the effectiveness of core stability exercises against which intervention? Active? passive? placebo? The lack of a well-defined comparator may have resulted in the found heterogeneity.

Line 134- In the methods section, specifically in the "data extraction" subsection, the data extraction tool is not described, nor is the data extracted from each study.

While the PRISMA checklist states that page 9 describes the methods for assessing certainty in the evidence, no such procedure was carried out. Considering the Joanna Briggs Institute and Cochrane guidelines for systematic reviews of effectiveness, assessing the certainty of the evidence is a mandatory step in this type of review, and provides the reader with an indication of the strength of the evidence produced by the systematic review. In agreement, authors should consider assessing the certainty of the results using the GRADE approach.

Results

Line 198/199- “As illustrated in Table 1, three studies were conducted in Iran, followed by Egypt

and Pakistan”. Please clarify this sentence. How many studies were conducted in Egypt and Pakistan?

Table 1. While categories for severity of intellectual disability are presented in the results section, the reader is not previously informed on which system this classification is based.

Table 3. Consider providing values for the intervention/ control effects.

According to relevant guidelines, a summary of findings table should be provided. Consider adding this table in the final of the results section accompanied by a grading of the certainty in the evidence.

Conclusion

In this section, consider providing information regarding the effectiveness of core stability exercises in improving each of the outcomes of interest in light of the certainty in the summarized evidence.

Reviewer #3: 1- number of articles in systematic review is very few 2- My major concern is that core stability exercises are not very useful and functional, nowadays. For example, functional training and postural re-education are very noticeable at the time being.

Reviewer #4: 1. Please kindly make the necessary corrections in line 69 (Studies have shown that individual's balance performance is associated with their core stability and not individuals balance).

2. Please kindly indicate the number/percentage of articles conducted in the various countries (Line 198)

Reviewer #5: Abstract

This abstract needs to be written more robustly. Firstly, it not clear what the rationale or significance of the study is from the back ground. There methods section is extremely brief and the results need to be presented more succinctly. Your conclusion appears exaggerated especially based on two studies for a narrative synthesis and four studies for a meta-analysis. I suggest that you consider being more measured in your conclusions based on limited data/evidence.

Introduction

Well done on your introduction.

Please consider these to improve the coherence.

Discuss the burden/epidemiological data around balance in your population of interest (ID). Based on this discuss the need for balance training amongst this population. Discuss the effects of balance training. Consider reducing the definitions and critique the existing literature around the phenomenon. You say there are studies existing in this area but these studies have not been systematically reviewed. Is that the rationale for your study? You need to be clear why there needs to be a systematic review in this area. Are these studies contrasting? Do you need to pull them together to firm up their conclusions although they all conclude on the same things? The biomechanics of core stability should be discussed robustly as well to link to the need to conduct this systematic review. Please consider these to improve the introduction.

Information sources and search strategy

What process will you use to ensure or assess inter-rater reliability of the data search, identification and extraction?

Inclusion criteria

Consider using a more robust and scientific approach to writing your inclusion criteria. Also, you need to provide an exclusion criteria.

Data extraction

What evidence informed your approach to data extraction? Again, this needs to be written more scientifically and backed by evidence.

Discussion

Your discussion is well written. However, I'm concerned about the firm conclusions you are drawing on effectiveness based on six studies. You need to consider whether the effect sizes are clinically meaningful also. Again, there should be a section that discusses the evidence for core stability exercises and its effect on balance more clearly. The strength of existing evidence should be discussed.

Reviewer #6: I am happy to read through this systematic review and metanalysis of the effects of core stability exercises on balance ability of children and adolescents with intellectual disabilities. To the best of my knowledge, the manuscript is sound and analysis well presented. I recommend that the work be accepted for publication if the necessary grammatical check is completed. Thank you

6. PLOS authors have the option to publish the peer review history of their article (what does this mean?). If published, this will include your full peer review and any attached files.

Reviewer #1: No

Reviewer #2: No

Reviewer #3: No

Reviewer #4: No

Reviewer #5: No

Reviewer #6: **Yes: **Nonso Asouzu

---

## [Author Response · Author response to Decision Letter 0]

22 Aug 2024

Dear reviewers:

Thank you for your letter and the reviewers’ comments on our manuscript entitled “Effects of core stability exercises on balance ability of children and adolescents with intellectual disabilities: A systematic review and meta-analysis” [PONE-D-24-21189]. Those comments are very helpful for revising and improving our paper. We have studied the comments carefully and made corrections which we hope to meet with approval. The main corrections are in the manuscript and the response to the reviewers’ comments are as follows (the replies are highlighted in bold).

Replies to the reviewers’ comments:

Reviewer #1: 

(1) I think it is a different field that can be reviewed. but I do not find it appropriate to publish it because I think that the existing studies are not sufficient for meta-analysis and the results will be misleading.

Response: Thank you for your comments. We have taken into account your suggestions and those of other reviewers and have endeavoured to improve our study. In addition, we have given due consideration to the certainty of the evidence and reached our conclusions with care. The positive attributes of core stability exercise, such as being low cost, and easy to implement, have made it an ideal option for balance improvement for individuals with disabilities, such as hearing impairment [1], visual impairment [2], and cerebral palsy [3]. Therefore, we hope to aggregate evidence from relevant trials to clearly illustrate the effectiveness of core stability exercises in improving balance performance for children and youth with ID. Our meta-analysis highlighted large effect size improvements over active control groups in both static and static-dynamic balance, though the results were not statistically significant. The limited number of eligible studies may reduce the reliability of results, but it emphasises the need for additional research in this domain. Future studies focusing on core stability exercises should include standardised clinical measures to assess balance and individualised approaches taken into account by clinical profiles. This will help develop core stability intervention strategies to support balance performance in children and adolescents with ID. Therefore, the results and perspectives provided in this systematic review and meta-analysis may provide an important contribution to understanding the impact of core stability exercises on improving balance and can have a broader impact on practice and research.

[1] Zarei H, Norasteh AA. Effects of proprioception and core stability training followed by detraining on balance performance in deaf male students: a three-arm randomized controlled trial. Somatosens Mot Res. 2023;1–9.

[2] Salar S, Karimizadeh Ardakani M, Lieberman LJ, Beach PS, Perreault M. The effects of balance and core stability training on postural control in people with visual impairment: A systematic review. Br J Vis Impair. 2022; 41(3): 528-541.

[3] Huang C, Chen Y, Chen G, Xie Y, Mo J, Li K, et al. Efficacy and safety of core stability training on gait of children with cerebral palsy. Med. 2020; 99(2): e18609.

Reviewer #2: 

In this manuscript, the authors summarise the available evidence on the effectiveness of core stability exercises in improving balance in children and adolescents with intellectual disabilities. The topic is clinically relevant and the results emphasise the need for more primary studies on the subject. However, some questions need to be properly addressed: 

(1) Abstract: Line 39/41- “Core stability exercises are effective in improving dynamic balance in children and adolescents with ID. However, no significant effects were found on the improvement in static and static-dynamic balance in this population”. Please consider clarifying the content of this sentence adding information about heterogeneity in the included sources of evidence, and the lack of significant improvements in comparison with the control group.

Response: We really appreciate your suggestions. We have revised the abstract section and reported more details in the discussion section as suggested. The revised text reads: “Core stability exercises appear to be a promising and acceptable intervention for the improvement of dynamic balance in children and adolescents with ID. Despite pooled analyses also highlighted improvements over active control groups in static and static-dynamic balance with large effect sizes, results were not statistically significant and should be interpreted with caution given the wide confidence intervals. The heterogeneity among the identified literature and the limited number of eligible studies may reduce the reliability of results, but it emphasises the need for additional research in this domain.” In addition, we have explained possible sources of heterogeneity in the discussion section (Lines 426-435 on pages 27-28 of the Manuscript).

(2) Methods-Line 126/127- According to the inclusion criteria, the average age of the participants should be < 18 years. However, the mean is a measure of central location, not indicate sample extremes. For example, in a hypothetical sample of 2 participants, if one is 30 years old and the other is 2 years old, the average will be 16 years old.

Response: Thank you very much for pointing out this problem. We revised the age of the inclusion criteria (Lines 150-152 on pages 8-9 of the Manuscript) after carefully reviewing the relevant systematic reviews and meta-analyses for populations of children and adolescents [1,2]. Meanwhile, we report the age range of participants included in the study in Table 1.

[1] McGarty A M, Downs S J, Melville C A, et al. A systematic review and meta‐analysis of interventions to increase physical activity in children and adolescents with intellectual disabilities. J Intellect Disabil Res. 2018; 62(4): 312-329.

[2] Zhou Y, Qi J. Effectiveness of Interventions on Improving Balance in Children and Adolescents With Hearing Impairment: A Systematic Review. Front Physiol. 2022; 13: 876974.

(3) Methods: Line 128/129- The comparator is not properly described. The review aims to assess the effectiveness of core stability exercises against which intervention? Active? passive? placebo? The lack of a well-defined comparator may have resulted in the found heterogeneity.

Response: Thank you for pointing this out. We have revised the details of the comparator. The revised text reads: “A standardised supervised conventional physical therapy program or any other training not focused on core exercising would be used as an active control group”.

(4) Methods: Line 134- In the methods section, specifically in the “data extraction” subsection, the data extraction tool is not described, nor is the data extracted from each study.

Response: Thank you very much for pointing out this problem. We have added details of the data extraction tool and the data extracted from each study (lines 176 on page 10 of the Manuscript).

(5) While the PRISMA checklist states that page 9 describes the methods for assessing certainty in the evidence, no such procedure was carried out. Considering the Joanna Briggs Institute and Cochrane guidelines for systematic reviews of effectiveness, assessing the certainty of the evidence is a mandatory step in this type of review, and provides the reader with an indication of the strength of the evidence produced by the systematic review. In agreement, authors should consider assessing the certainty of the results using the GRADE approach.

Response: We are very grateful for your advice. We have added details on assessing the certainty of the body of evidence using the GRADE approach (lines 202 on page 11 of the Manuscript).

(6) Results: Line 198/199- “As illustrated in Table 1, three studies were conducted in Iran, followed by Egypt and Pakistan”. Please clarify this sentence. How many studies were conducted in Egypt and Pakistan?

Response: Thank you very much for your advice, we have added relevant details. The revised text reads: “As illustrated in Table 1, three studies (50%) were conducted in Iran, followed by Egypt (n = 2, 33.3%) and Pakistan (n = 1, 16.7%)”.

(7) Table 1. While categories for severity of intellectual disability are presented in the results section, the reader is not previously informed on which system this classification is based.

Response: Thank you very much for your advice. Based on the inclusion of the studies in this review, we adopted the Diagnostic and Statistical Manual of Mental Disorders (4th edition) [1] for categorizing the severity of intellectual disability: Mild (IQ level 50-55 to approximately 70), Moderate (IQ level 35-40 to 50-55), Severe (IQ level 20-25 to 35-40) and Profound (IQ level below 20 or 25). We have added information about the classification criteria in the results section (lines 267-271 on page 14 of the Manuscript).

[1] American Psychiatric Association. Diagnostic and Statistical Manual of Mental Disorders: DSM-IV. 4th ed. Washington, DC: American Psychiatric Association; 1994.

(8) Table 3. Consider providing values for the intervention/ control effects.

Response: Thank you very much for your advice. We have added values for the intervention/ control effects in Table 3.

(9) Results: According to relevant guidelines, a summary of findings table should be provided. Consider adding this table in the final of the results section accompanied by a grading of the certainty in the evidence.

Response: Thank you very much for pointing out this problem. We have added the grading results suggested by the GRADE system to Table 4, along with a description of the evidence grading results in the results section (lines 330 on page 22 of the Manuscript).

(10) Conclusion: In this section, consider providing information regarding the effectiveness of core stability exercises in improving each of the outcomes of interest in light of the certainty in the summarized evidence.

Response: Thank you very much for pointing out this problem. We have added information about the effectiveness of core stability exercises in improving each of the outcomes of interest in the conclusions section. The revised text reads: “This review indicates that core stability exercises may be a promising means to improving dynamic balance in children and adolescents with ID, suggesting the potential feasibility of incorporating it into an important part of the physical and social rehabilitation program for these children. In addition, despite the quantitative results highlighting improvements over active control groups in static and static-dynamic balance with large effect sizes, results were not statistically significant and should be interpreted with caution, given the wide confidence intervals. Therefore, further rigorous studies are needed to strengthen the evidence in this area, given the quality of the underlying evidence base is currently low.”

Reviewer #3: 

(1) number of articles in systematic review is very few 

Response: We agree with your valuable comments. In recent years, several studies [1,2] have reported that core stability exercises improve the balance ability of individuals with disabilities. Therefore, we hope to aggregate evidence from relevant trials to clearly illustrate the effectiveness of core stability exercises in improving balance performance for children and adolescents with ID. Despite the limited number of eligible studies that may reduce the reliability of results, it emphasises the need for additional research in this domain. Future studies and limitations are discussed in this study. Therefore, the results and perspectives provided in this systematic review and meta-analysis may provide an important contribution to understanding the impact of core stability exercises on improving balance and can have a broader impact on practice and research.

[1] Salar S, Ardakani M, Lieberman L, Beach P, Perreault M. The effects of balance and core stability training on postural control in people with visual impairment: A systematic review. Br J Vis Impair. 2022;026461962210772. 

[2] Zarei H, Norasteh AA. Effects of proprioception and core stability training followed by detraining on balance performance in deaf male students: a three-arm randomized controlled trial. Somatosens Mot Res. 2023; 40(2): 47–55. 

(2) My major concern is that core stability exercises are not very useful and functional, nowadays. For example, functional training and postural re-education are very noticeable at the time being.

Response: We are very grateful that you have recommended two very valuable interventions. Currently, functional training and postural re-education have also acquired our attention. Some studies [1,2] have shown that functional training is effective in improving balance in individuals with intellectual disabilities, but some rigorous RCTs seem to be lacking. Several studies have shown that postural re-education appears to be an effective exercise rehabilitation program to improve neck and lower back pain [3][4]. However, postural reeducation seems to have little research on improving balance ability, and has not been widely used in children and adolescents. Core stability exercises are characterised by their low cost, ease of implementation, and alignment with the physical and mental capabilities of children with intellectual disabilities [5]. Therefore, it is necessary to demonstrate the effectiveness of core stability exercises in improving balance performance in children with intellectual disabilities and to apply them in physical and social rehabilitation programs.

[1] Mikolajczyk E, Jankowicz-Szymanska A. The effect of dual-task functional exercises on postural balance in adolescents with intellectual disability – a preliminary report. Disabil Rehabil. 2014; 37(16):1484–9.

[2] Mikolajczyk E, Jankowicz-Szymanska A. Does extending the dual-task functional exercises workout improve postural balance in individuals with ID? Res Dev Disabil. 2015; 38: 84–91. 

[3] Fernandes TM, Méndez-Sánchez R, Puente-González AS, Martín-Vallejo FJ, Falla D, Carolina VC. A randomized controlled trial on the effects of “Global Postural Re-education” versus neck specific exercise on pain, disability, postural control, and neuromuscular features in women with chronic non-specific neck pain. Eur J Phys Rehabil Med. 2023; 59(1): 42.

[4] Cavalcanti IF, Antonino GB, Monte-Silva KK do, Guerino MR, Ferreira AP de L, das Graças Rodrigues de Araújo M. Global Postural Re-education in non-specific neck and low back pain treatment: A pilot study. J Back Musculoskelet Rehabil. 2020; 33(5): 823–8.

[5] Ghaeeni S, Bahari Z, Khazaei AA. Effect of Core Stability Training on Static Balance of the Children with Down Syndrome. Phys Ther. 2015; 5(1): 49-53.

Reviewer #4:

(1) Please kindly make the necessary corrections in line 69 (Studies have shown that individual’s balance performance is associated with their core stability and not individuals balance).

Response: Thank you very much for your valuable comment. We have made the necessary changes to the sentence (lines 88 on page 5 of the Manuscript).

(2) Please kindly indicate the number/percentage of articles conducted in the various countries (Line 198).

Response: Thank you very much for pointing out this problem. We have revised this section. The revised text reads: “As illustrated in Table 1, three studies were conducted in Iran, followed by Egypt (n = 2, 33.3%) and Pakistan (n = 1, 16.7%)”.

Reviewer #5:

(1) Abstract: This abstract needs to be written more robustly. Firstly, it not clear what the rationale or significance of the study is from the background. There methods section is extremely brief and the results need to be presented more succinctly. Your conclusion appears exaggerated especially based on two studies for a narrative synthesis and four studies for a meta-analysis. I suggest that you consider being more measured in your conclusions based on limited data/evidence. 

Response: Thank you very much for your valuable comment. We have rewritten the abstract section to make it more rigorous (lines 14-46 on pages 2-3 of the Manuscript).

(2) Introduction: Well done on your introduction. Please consider these to improve the coherence. Discuss the burden/epidemiological data around balance in your population of interest (ID)

---

## [Decision Letter · Decision Letter 1]

23 Sep 2024

PONE-D-24-21189R1Effects of core stability exercises on balance ability of children and adolescents with intellectual disabilities: A systematic review and meta-analysisPLOS ONE

Dear Dr. xu,

Thank you for submitting your manuscript to PLOS ONE. After careful consideration, we feel that it has merit but does not fully meet PLOS ONE’s publication criteria as it currently stands. Therefore, we invite you to submit a revised version of the manuscript that addresses the points raised during the review process.

We look forward to receiving your revised manuscript.

Kind regards,

Domiziano Tarantino, MD

Academic Editor

PLOS ONE

Journal Requirements:

Reviewers' comments:

Reviewer's Responses to Questions

**Comments to the Author**

1. If the authors have adequately addressed your comments raised in a previous round of review and you feel that this manuscript is now acceptable for publication, you may indicate that here to bypass the “Comments to the Author” section, enter your conflict of interest statement in the “Confidential to Editor” section, and submit your "Accept" recommendation.

Reviewer #2: All comments have been addressed

Reviewer #5: All comments have been addressed

2. Is the manuscript technically sound, and do the data support the conclusions?

Reviewer #2: Partly

Reviewer #5: Yes

3. Has the statistical analysis been performed appropriately and rigorously? 

Reviewer #2: Yes

Reviewer #5: Yes

4. Have the authors made all data underlying the findings in their manuscript fully available?

Reviewer #2: Yes

Reviewer #5: Yes

5. Is the manuscript presented in an intelligible fashion and written in standard English?

Reviewer #2: No

Reviewer #5: Yes

6. Review Comments to the Author

**Reviewer #2:** I congratulate the authors as the new version of the manuscript presents substantial improvements.

All my initial concerns were addressed by the authors in the first review of the manuscript.

The conclusion described in the abstract ‘Core stability exercises appear to be a promising and acceptable intervention for the improvement of dynamic balance in children and adolescents with ID’ seems abusive to me, considering the scarcity of studies included, the results of the studies and the certainty of the evidence synthesised.

In order to be published, the manuscript must be reviewed idiomatically and scientifically in terms of writing style.

**Reviewer #5:** Well done on putting this manuscript together.

You have addressed all my comments very well. All the best!

7. PLOS authors have the option to publish the peer review history of their article (what does this mean?). If published, this will include your full peer review and any attached files.

Reviewer #2: No

Reviewer #5: No

---

## [Author Response · Author response to Decision Letter 1]

4 Oct 2024

Dear reviewers:

Thank you once again for your constructive comments on our revised manuscript titled “Effects of core stability exercises on balance ability of children and adolescents with intellectual disabilities: A systematic review and meta-analysis” [PONE-D-24-21189R1]. Your insights have been invaluable in refining our work, and we have carefully addressed each of your suggestions in this second round of revisions. The main corrections are in the manuscript and the response to the reviewers’ comments are as follows (the replies are highlighted in bold).

Replies to the reviewers’ comments:

Reviewer #2: I congratulate the authors as the new version of the manuscript presents substantial improvements. All my initial concerns were addressed by the authors in the first review of the manuscript. The conclusion described in the abstract ‘Core stability exercises appear to be a promising and acceptable intervention for the improvement of dynamic balance in children and adolescents with ID’ seems abusive to me, considering the scarcity of studies included, the results of the studies and the certainty of the evidence synthesised. In order to be published, the manuscript must be reviewed idiomatically and scientifically in terms of writing style.

Response: Thank you for your positive comments. We have revised the conclusion section of the abstract. The revised text reads: “Core stability exercises may improve dynamic balance in children and adolescents with ID, but given the scarcity of studies included, definitive conclusions cannot yet be drawn.” (Page 3, lines 39-41). Our manuscript has been meticulously edited by professional editors from AJE’s editing team to ensure correct English language, grammar, punctuation, spelling, and overall style (you can verify this using the verification code 5833-2A66-21B7-AB08-9768 on the AJE website).

Reviewer #5: 

Well done on putting this manuscript together.

You have addressed all my comments very well. All the best 

Response: Thank you very much for your encouragement. We are glad to hear that we have effectively addressed your comments. Your thorough review and constructive feedback have been instrumental in improving our manuscript. We appreciate your time and effort and wish you all the best as well.

Kind regards

Sincerely,

Junjiie Zhou (The First Author)

E-mail: zhoujunjie@zjnu.edu.cn

Wenhong Xu (Corresponding author)

E-mail: xuwenhong@zjnu.edu.cn

04 October, 2024

College of Physical Education and Health Sciences

Zhejiang Normal University

Jinhua, China

---

## [Decision Letter · Decision Letter 2]

14 Nov 2024

Effects of core stability exercises on balance ability of children and adolescents with intellectual disabilities: A systematic review and meta-analysis

PONE-D-24-21189R2

Dear Dr. Xu,

We’re pleased to inform you that your manuscript has been judged scientifically suitable for publication and will be formally accepted for publication once it meets all outstanding technical requirements.

Kind regards,

Domiziano Tarantino, MD

Academic Editor

PLOS ONE

Additional Editor Comments (optional):

Reviewers' comments:

Reviewer's Responses to Questions

**Comments to the Author**

1. If the authors have adequately addressed your comments raised in a previous round of review and you feel that this manuscript is now acceptable for publication, you may indicate that here to bypass the “Comments to the Author” section, enter your conflict of interest statement in the “Confidential to Editor” section, and submit your "Accept" recommendation.

Reviewer #2: All comments have been addressed

2. Is the manuscript technically sound, and do the data support the conclusions?

Reviewer #2: Yes

3. Has the statistical analysis been performed appropriately and rigorously? 

Reviewer #2: Yes

4. Have the authors made all data underlying the findings in their manuscript fully available?

Reviewer #2: Yes

5. Is the manuscript presented in an intelligible fashion and written in standard English?

Reviewer #2: Yes

6. Review Comments to the Author

Reviewer #2: I congratulate the authors on the final version of the manuscript. All my initial concerns have been properly addressed.

7. PLOS authors have the option to publish the peer review history of their article (what does this mean?). If published, this will include your full peer review and any attached files.

Reviewer #2: No

---

## [Editor Report · Acceptance letter]

9 Dec 2024

PONE-D-24-21189R2 

PLOS ONE

Dear Dr. xu, 

I'm pleased to inform you that your manuscript has been deemed suitable for publication in PLOS ONE. Congratulations! Your manuscript is now being handed over to our production team.

Kind regards, 

on behalf of

Dr. Domiziano Tarantino 

Academic Editor

PLOS ONE